# A spinoparabrachial circuit defined by Tacr1 expression drives pain

**Arnab Barik[1†]\*, Anupama Sathyamurthy[2†], James Thompson[1], Mathew Seltzer[1], Ariel Levine[2], Alexander Chesler[1,2]\***

[1]National Center for Complementary and Integrative Health, National Institutes of Health, Bethesda, United States; [2]National Institute of Neurological Disorders and Stroke, National Institutes of Health, Bethesda, United States

**Abstract** Painful stimuli evoke a mixture of sensations, negative emotions and behaviors. These myriad effects are thought to be produced by parallel ascending circuits working in combination. Here, we describe a pathway from spinal cord to brain for ongoing pain. Activation of a subset of spinal neurons expressing Tacr1 evokes a full repertoire of somatotopically directed pain-related behaviors in the absence of noxious input. Tacr1 projection neurons (expressing NKR1) target a tiny cluster of neurons in the superior lateral parabrachial nucleus (PBN-SL). We show that these neurons, which also express Tacr1 (PBN-SL[Tacr1]), are responsive to sustained but not acute noxious stimuli. Activation of PBN-SL[Tacr1] neurons alone did not trigger pain responses but instead served to dramatically heighten nocifensive behaviors and suppress itch. Remarkably, mice with silenced PBN-SL[Tacr1] neurons ignored long-lasting noxious stimuli. Together, these data reveal new details about this spinoparabrachial pathway and its key role in the sensation of ongoing pain.

**\*For correspondence:**
arnabbarik@iisc.ac.in (AB);
alexander.chesler@nih.gov (AC)

**Present address:** [†]Center for Neuroscience, Indian Institute of Science, Bengaluru, India

**Competing interests:** The authors declare that no competing interests exist.

## Introduction

Somatosensory input to the spinal cord provides information about both the external environment and internal state, driving reflex responses and complex perceptual experiences including the sensation of pain. Nociception provides animals with crucial protection from mechanical, thermal, and chemical threats, but many other types of sensory input can become painful after injury, during inflammation, or in disease. An extensive body of work has explored how networks of cells in the dorsal spinal cord process noxious versus innocuous sensory input and the context-dependent plasticity in these circuits that results in pain. Many different classes of excitatory and inhibitory interneurons have been shown to be required for processing nociceptive input and the transformation of normally innocuous signals into pain. These include cells expressing select combinations of neuropeptides and their receptors, including Substance P (encoded by *Tac1*) and its receptor NKR1 (encoded by *Tacr1*). Ultimately, these spinal circuits activate projection neurons that transmit the highly processed input to the brain.

Spinal projection neurons are sparse and heterogenous in their gene expression and their anatomical location (*Koch et al., 2018*; *Peirs and Seal, 2016*; *Todd, 2010*) but again include cells expressing both Substance P and NKR1 (*Huang et al., 2019*; *Chiang et al., 2020* and *Choi et al., 2020*). For example, one group resides in the most superficial part of the dorsal horn (Lamina I) and is thought to receive input from peripheral nociceptors, itch neurons, and thermoreceptors (*Bester et al., 2000*; *Wercberger et al., 2020*; *Wercberger and Basbaum, 2019*). A second set of projection neurons is located in the deep dorsal horn (Laminae II-VI) where they integrate noxious and low-threshold sensory inputs (*Wall, 1967*; *Wercberger and Basbaum, 2019*; *Willis, 1983*). Both groups of spinal projection neurons target multiple sites in the brainstem and thalamus, with each site proposed to have unique roles in driving different aspects of sensation, learning and behavior

(*Choi et al., 2020*, *Huang et al., 2019*; *Rpc, 1986*; *Todd, 2010*). The prevailing view is that these parallel outputs combine to elicit the full pain experience. If true, then direct activation of the correct ensemble of projection neurons should produce behaviors indistinguishable from those evoked by actual noxious stimuli. By contrast, activating or silencing individual central targets might be expected to selectively affect particular aspects of pain and potentially reveal substrates that encode specific attributes of this complex state.

The lateral parabrachial nucleus (lPBN), a brainstem nucleus of a few thousand neurons in mice, receives nociceptive inputs from the projection neurons in the dorsal spinal cord (*Todd, 2010*) and plays a central role in pain. Two subregions of the lPBN have been defined by their distinctive gene expression patterns as well as their forebrain projections. Neurons in the dorsal PBN (dPBN) expressing dynorphin (PDyn) project predominantly to the periaqueductal gray (PAG) and ventromedial hypothalamus (VMH), while external lateral parabrachial neurons (elPBN) expressing the neuropeptides CGRP or Substance P project to the central amygdala (CeA), bed nucleus stria terminalis (BNST), insular cortex (INS), and the medullary formation (MdD) (*Rodriguez et al., 2017*; *Palmiter, 2018*; *Barik et al., 2018*). Recent evidence suggests that these different output pathways have differing functional specifications. For example, CGRP projections to the amygdala appear particularly important for somatic and visceral distress (*Carter et al., 2013*; *Han et al., 2015*; *Campos et al., 2018*; *Bowen et al., 2020*) and also play a role in affective touch (*Choi et al., 2020*). By contrast, Tac1-positive neurons projecting to the MdD (*Barik et al., 2018*) participate in nocifensive responses, whereas the Pdyn neurons in the dPBN are important for sensing gut stretch (*Kim et al., 2020*) and forming aversive memories (*Chiang et al., 2020*).

Here, we studied projections from the spinal cord to the PBN and discovered a population of lPBN neurons expressing Tacr1 that receive input from Tacr1-expressing spinal projection neurons. We demonstrate that this circuit is both necessary and sufficient for the heightened behavioral responses observed during ongoing noxious stimulation. Direct activation of the spinal Tacr1 neurons causes striking behaviors that closely match pain responses. Interestingly, activation of their parabrachial Tacr1 targets demonstrates that this branch of the ascending pathway is responsible for controlling the magnitude of the behavioral response and appears to be required for affective aspects of pain sensation. Our data complement and extend recent reports highlighting the roles of both spinal (*Choi et al., 2020*) and parabrachial (*Deng et al., 2020*) Tacr1 neurons as important drivers of pain behavior.

## Results

Tacr1 has been widely used as a marker for projection neurons in the spinal cord (*Todd, 2010*); however, not all projection neurons are marked by Tacr1 (*Choi et al., 2020*) and interneurons also express this gene (*Sathyamurthy et al., 2020*; *Sathyamurthy et al., 2018*) Nonetheless, the spinal-$^{Tacr1}$ neurons are well known to play a crucial role in normal nociceptive behaviors (*Mantyh et al., 1997*). We initially characterized mice where *Tacr1$^{Cre}$* was used to drive GFP-expression to assess the proportions of projection neurons and interneurons targeted (*Figure 1—figure supplement 2A, B*). Our results confirmed that *Tacr1$^{Cre}$* mediates recombination both in the superficial and deep dorsal horn with about 40% of targeted neurons having soma diameters of projection neurons (*Al Ghamdi et al., 2009*). Next, we examined how activation of spinal$^{Tacr1}$ neurons contributes to pain sensation by directly stimulating a small circumscribed group of these cells using chemogenetics. *Tacr1$^{Cre}$* mice were transduced with an AAV encoding the excitatory DREADD receptor hM3Dq fused to an mCherry reporter (*Barik et al., 2018*; *Krashes et al., 2011*) on the right side of superficial lumbar dorsal horn (*Figure 1A*). As expected, immunohistochemical analysis confirmed expression by neurons in Lamina I and Laminae III-V on the side ipsilateral to the injection location (L4; *Figure 1B*). Importantly, administration of the hM3Dq ligand clozapine-N-oxide (CNO) resulted in widespread Fos expression on the injected side (*Figure 1B*), validating this stimulation paradigm (see *Figure 1—figure supplement 1A, B* for quantitation). Our results showed that both hM3Dq-mCherry positive and negative neurons express Fos after CNO delivery. A likely explanation is that CNO activates Tacr1+ neurons, and that Tacr1+ neurons in turn activate other non-Tacr1 neurons.

Each side of a lumbar spinal cord segment receives sensory inputs from the ipsilateral hindlimb (*Basbaum et al., 2009*). Within minutes of CNO application, animals with spinal$^{Tacr1}$ neurons expressing hM3Dq began spontaneously lifting, shaking and licking the hindlimb ipsilateral to the

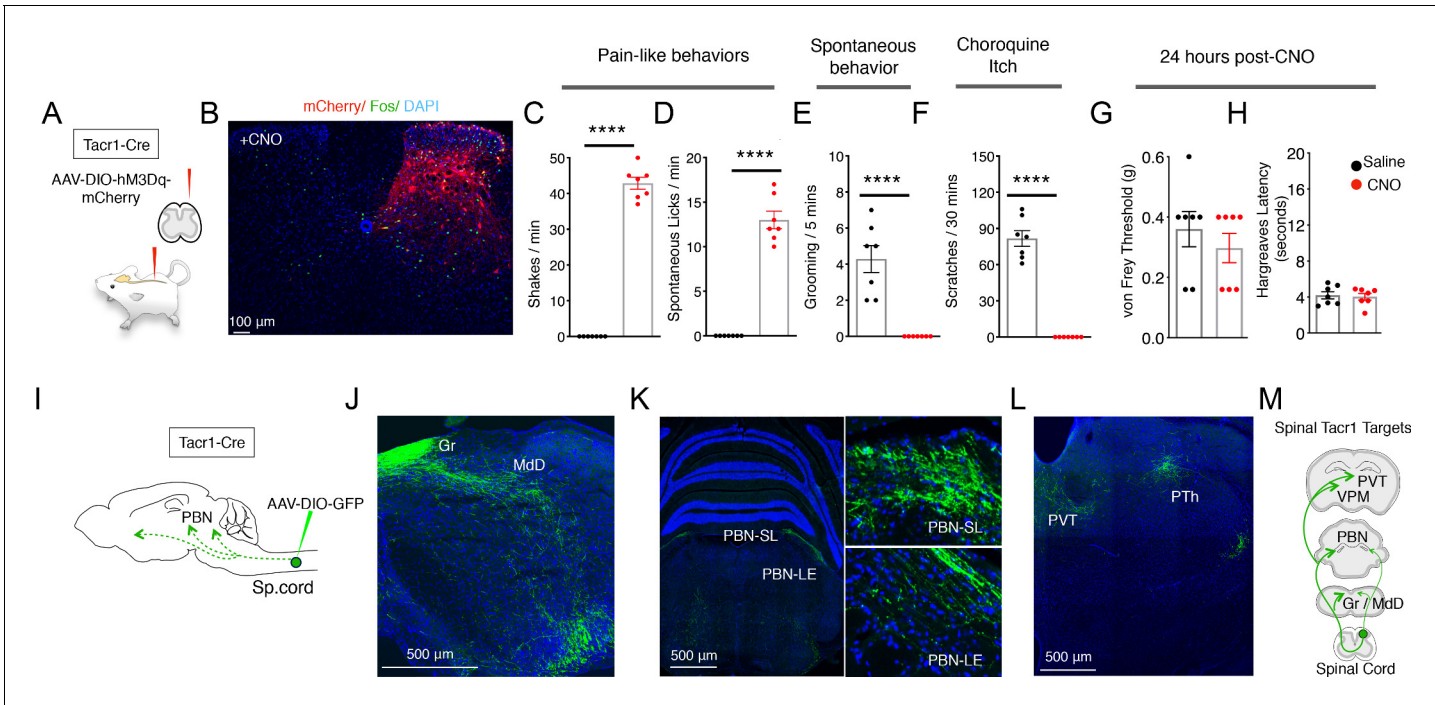

**Figure 1.** Activation of Tacr1+ projection neurons causes fictive pain. (**A**) A cartoon depicting the strategy to stimulate Tacr1-positive spinal neurons. An AAV viral vector encoding a Cre-dependent stimulating DREADD receptor (AAV-DIO-hM3Dq-mCherry) was injected into the lumbar spinal cord of Tacr1-Cre mice to transduce a localized subset of Tacr1-neurons. (**B**) A confocal image of a coronal section of lumbar spinal cord from an injected Tacr1-Cre mouse shows that the hM3Dq expression (red) is limited to neurons in the ipsilateral superficial and deep dorsal horn. CNO administration results in induction of the activity-dependent gene Fos (green). Scale = 100 µm (**C–H**) Bar graphs showing quantified CNO-induced behavioral responses; saline control, black; CNO, red; n = 7 mice/group; ****p≤0.0001, unpaired two-tailed t test. CNO-induced typical pain responses: ipsilateral hindpaw shaking (C; p<0.0001) and licking (D; p=<0.0001), suppressed normal grooming (E; p=<0.0001) and itch responses to chloroquine injection in a different somatotopic site (F; p=<0.0001). As expected, behavioral changes were fully reversible after 24 hr (**G–H**): shown are threshold responses to punctate touch (von Frey in grams; G; p=0.4245) and latencies to respond to radiant heat (Hargreaves test; H; p=0.7640). (**I**) A cartoon depicting the strategy to label the ascending projections of a small, circumscribed group of Tacr1+ projection neurons. An AAV viral vector encoding a Cre-dependent GFP marker (AAV-DIO-GFP) was injected into the L4 region of the lumbar spinal cord of Tacr1-Cre mice. (**J–L**) Coronal sections showing major targets of Spinal$^{Tacr1}$ neurons. In the brainstem, dense GFP-labeled projections were detected in (**J**) the gracile nucleus (Gr), reticular formation (MdD) and (**K**) two regions of the parabrachial nucleus (PBN-EL and PBN-SL). GFP-positive Spinal$^{Tacr1}$ projections are also observed in midline brain regions: the paraventricular thalamus (PVT) and posterior thalamus (PTh). (**M**) A cartoon summarizing the ascending pathway of spinal Tacr1-expressing projection neurons. Green = GFP; Blue = DAPI. Scales = 500 µm.

The online version of this article includes the following figure supplement(s) for figure 1:

**Figure supplement 1.** Validation of chemogenetic strategy to stimulate Spinal$^{Tacr1}$ neurons.

**Figure supplement 2.** Tacr1$^{Cre}$ labels both projection neurons and interneurons.

**Figure supplement 3.** Fos induction by Chloroquine injection into the midline nape of the neck is not suppressed by chemogenetic activation of Spinal$^{Tacr1}$ neurons in lumbar region.

**Figure supplement 4.** Tac1 and Tacr1 are co-expressed in nociceptive spinal projection neurons.

AAV injection (*Figure 1C and D*; *Videos 1* and *2*). These responses were highly reminiscent of those caused by extremely noxious and persistent stimuli such as chemical irritants (*Kwan et al., 2006*), lasted for over an hour and suppressed normal homecage behaviors (e.g. grooming; *Figure 1E*). Remarkably, animals on CNO became so focused on the DREADD-induced 'fictive' pain associated with the ipsilateral hindlimb that they ignored other highly salient stimuli. In particular, they were completely unresponsive to chloroquine, a potent puritogen (*Han et al., 2013*; *Mishra and Hoon, 2013*) that normally causes robust and long-lasting scratching (*Figure 1F*). Notably, chloroquine was applied to the nape of the neck, a completely different body area, and still induced Fos expression in the cervical spinal cord (*Figure 1—figure supplement 3*). Thus, in addition to directing nocifensive behavior to a somatotopically conscribed area, the activation of a small group of lumbar

Chemogenetic Activation of Tacr1$^{Spinal\ Cord}$ Neurons

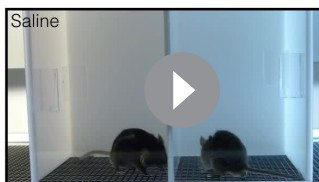

**Video 1.** Spontaneous behavior of Tacr1$^{Cre}$ mice expressing hM3Dq in the lumbar region of the spinal cord after injection of Saline. One minute of video recording of two mice showing normal baseline behaviors in testing chambers after habituation and saline injection.

https://elifesciences.org/articles/61135#video1

spinal$^{Tacr1}$ neurons produced the well-known phenomenon of itch suppression by pain (*Ikoma et al., 2006*). Moreover, our data strongly suggest that itch suppression by pain must involve a central mechanism since peripheral signaling persists. Finally, as expected, the effects of CNO were fully reversible and did not cause any long-lasting sensitization (*Figure 1G and H*).

We were surprised that artificial stimulation of the spinal$^{Tacr1}$ neurons so accurately phenocopied behaviors caused by genuine noxious stimulation since various types of dorsal horn projection neurons are considered necessary for pain behavior (*Koch et al., 2018*; *Todd, 2010*). Interestingly, a recent study showed that ablating spinal neurons expressing *Tac1*, the gene encoding the neuropeptide Substance P, significantly impaired licking responses to sustained noxious stimuli (*Huang et al., 2019*). In the spinal cord, *Tac1* is also broadly expressed both in projection neurons and interneurons (*Gutierrez-Mecinas et al., 2017*; *Huang et al., 2019*). Our in-situ hybridization data (*Figure 1—figure supplement 4A, B*)

revealed cells with high levels of *Tacr1* transcripts in the superficial dorsal horn almost always co-express *Tac1*. We reasoned that activation of spinal$^{Tac1}$ neurons might produce similar behavioral effects as above (*Figure 1*). As predicted, chemogenetic stimulation of lumbar spinal$^{Tac1}$ neurons (*Figure 1—figure supplement 4C, E*) resulted in robust nocifensive behaviors directed to the ipsilateral hindlimb while at the same time suppressing grooming (*Figure 1—figure supplement 4F*) and itch responses (*Figure 1—figure supplement 4G*). There were minor differences in the behaviors with stimulation of *Tac1$^{Cre}$* animals being more robust than *Tacr1$^{Cre}$* mice (compare *Videos 2* and *3*), perhaps in part due to activation of *Tac1* (Substance P but not NKR1) expressing sensory neurons in the dorsal horn and dorsal root ganglion (*Figure 1—figure supplement 4H*). These data support the existence of key populations of dorsal horn neurons mediating pain responses that can be

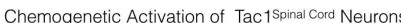

Chemogenetic Activation of Tacr1$^{Spinal\ Cord}$ Neurons

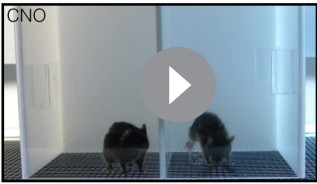

**Video 2.** Spontaneous behavior of Tacr1$^{Cre}$ mice expressing hM3Dq in the lumbar region of the spinal cord after injection of CNO. One minute of video recording of two mice showing pronounced nocifensive behaviors directed to the ipsilateral hindlimb in testing chambers approximately 30 min after CNO administration.

https://elifesciences.org/articles/61135#video2

Chemogenetic Activation of Tac1$^{Spinal\ Cord}$ Neurons

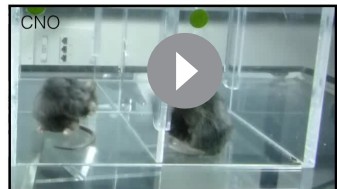

**Video 3.** Spontaneous behavior of Tac1$^{Cre}$ mice expressing hM3Dq in the lumbar region of the spinal cord after injection of CNO. One minute of video recording of two mice showing pronounced nocifensive behaviors directed to the ipsilateral hindlimb ipsilateral in testing chambers approximately 30 min after CNO administration (approximately 20 min post-injection).

https://elifesciences.org/articles/61135#video3

defined by the expression of *Tac1* and *Tacr1*. Importantly, our results demonstrate that activity of the spinal cells expressing these genes carries the information needed to elicit a pain sensation localized to a specific body area.

Pain has many dimensions including location, quality, intensity, and duration. It also affects physiology, emotion, attention, and memory (*Bushnell et al., 2013*). Output from the dorsal horn of the spinal cord targets multiple brain areas that work in concert to generate this complex state (*Basbaum et al., 2009*; *Todd, 2010*). We transduced lumbar spinal$^{Tacr1}$ neurons with an AAV-DIO-GFP to visualize their supraspinal projections (*Figure 1I*). Consistent with their ability to coordinate a full-blown pain response, anterograde tracing revealed GFP$^+$ axons in many regions of the brain (*Figure 1J-M*). We focused on the PBN because of its importance in pain behavior (*Barik et al., 2018*; *Chiang et al., 2020*; *Huang et al., 2019*). Outputs from two PBN areas, the lateral external and dorsal region, are known to have diverse impacts on pain responses, threat learning and escape behaviors (*Bowen et al., 2020*; *Campos et al., 2018*; *Chiang et al., 2020*; *Han et al., 2015*). Although GFP$^+$ axons targeted both these regions, the highest density of spinal$^{Tacr1}$ projections was at the very top of the superior lateral PBN (PBN-SL; *Figure 1K*), a small and anatomically distinct area with unknown function and is sometimes referred to as the internal lateral PBN (*Choi et al., 2020*). As expected, chemogenetic activation of spinal$^{Tacr1}$ (*Figure 2A*) and spinal$^{Tac1}$ (*Figure 1—figure supplement 4I*) neurons induced robust Fos expression in the PBN-SL.

Which cells in the PBN-SL receive the major input from the spinal dorsal horn and how do they contribute to pain sensation? To answer these questions, we sought molecular markers that could be used to genetically manipulate PBN-SL neurons so we could study their anatomy and function. Since both the neuropeptide (Tac1) and one of its receptors (Tacr1) were expressed by the spinal projection neurons, (*Figure 1—figure supplement 4A*) we predicted that tachykinin signaling molecules might mark this sensory circuit. Indeed, multiplex fluorescent in-situ hybridization revealed Tacr1 to be highly expressed in a tight cluster of excitatory cells in the PBN-SL (*Figure 2B,C* and *Figure 2—figure supplement 1A*). The expression of Tacr1 in both the spinal and PBN-SL neurons meant their anatomical connectivity could be probed by differentially transducing each region with AAVs in the same Tacr1$^{Cre}$ animal (*Figure 2D*). We labeled the presynaptic specializations of spinal-$^{Tacr1}$ axons with Synaptophysin-GFP and the postsynaptic specializations of the PBN-SL$^{Tacr1}$ dendrites and soma using PSD95-tagRFP (*Figure 2E*). Near super-resolution imaging of single Z planes from stained tissue sections from the PBN-SL showed very close apposition of red and green varicosities indicative of functional synapses (*Figure 2F,G*).

Despite its compact structure, the PBN appears central to many different sensory pathways with the various anatomic regions selectively targeting distinct higher centers and controlling different aspects of visceral sensation (*Bernard et al., 1994*), thermoregulation (*Yahiro et al., 2017*; *Yang et al., 2020*), itch (*Campos et al., 2018*; *Mu et al., 2017*), and pain (*Bernard and Besson, 1990*; *Campos et al., 2018*; *Menendez et al., 1996*). Therefore, we next used anterograde tracing to determine the projections of PBN-SL$^{Tacr1}$ neurons by targeting the PBN of Tacr1$^{Cre}$ animals with an AAV-DIO-tdTomato (*Figure 2H,I*). Two forebrain regions were prominently innervated by PBN-SL$^{Tacr1}$ neurons (*Figure 2J*): (1) the midline thalamus (MTh) and (2) the lateral hypothalamus/ parasubthalamic nucleus (LH/PSTN). Both these regions appear to be selective targets of PBN-SL excitatory projections that are not innervated by other lateral PBN subnuclei (*Barik et al., 2018*). Moreover, expression of synaptophysin-GFP in PBN-SL$^{Tacr1}$ neurons provided evidence for presynaptic terminal specializations in both areas (*Figure 2—figure supplement 2*).

Do the same neurons target both MTh and LH/PSTN? Or are there two distinct types of PBN-SL$^{Tacr1}$ neurons with different higher brain targets and potential roles? To answer these questions, we adopted an intersectional strategy where we labeled PBN-SL$^{Tacr1}$ neurons projecting to the LH/PSTh. Our approach used a Cre-dependent mCherry to label the cell bodies of Tacr1-expressing neurons in the PBN. We also injected the PSTN with retroAAV-FlpO and used a PBN injected Flp-dependent synaptophysin-GFP (*Sathyamurthy et al., 2020*) to label the synaptic termini of these cells (*Figure 3A,B*; see Materials and methods for details). We confirmed that the PBN-SL neurons expressed both labels (*Figure 3A*) meaning that PBN-SL$^{Tacr1}$ neurons project to the LH/PSTN, validating our approach. Importantly, Synaptophysin-GFP puncta were also abundant in the MTh (*Figure 3B*) demonstrating collateralization of PBN cells that we presume express *Tacr1*. Together, these anatomical experiments reveal two previously unappreciated targets of PBN-SL$^{Tacr1}$ neurons in

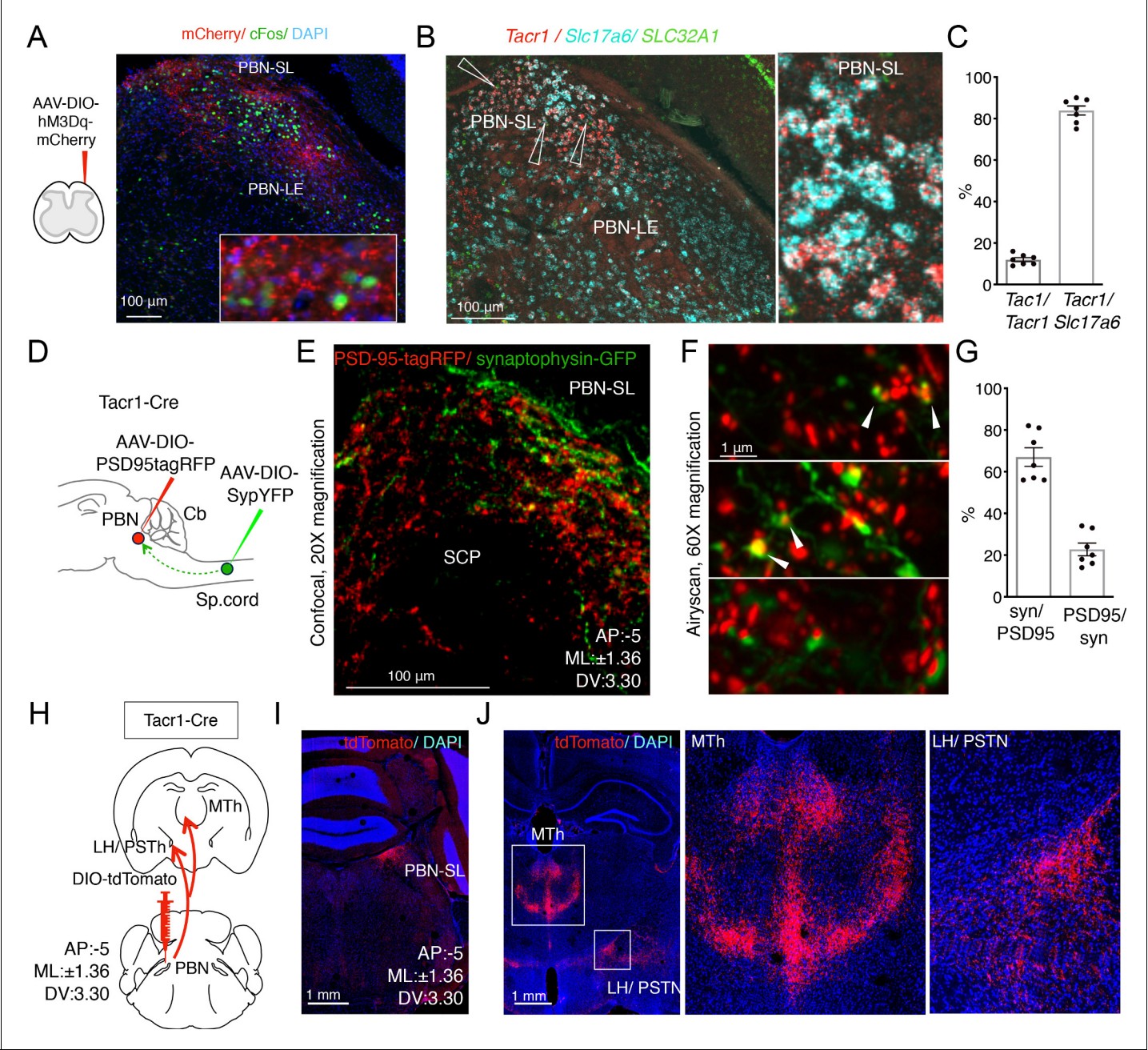

**Figure 2.** Anatomical organization of a Tacr1-defined spinoparabrachial circuit. (**A**) Spinal^Tacr1 neurons were transduced to express hM3Dq-mCherry (cartoon, left); typical confocal image of a coronal section of the PBN showing mCherry-positive Spinal^Tacr1 projections (right panel, red) in two regions of the PBN (PBN-EL and PBN-SL). Application of CNO resulted in Fos induction throughout the PBN (green). The nuclear stain DAPI (blue) highlights the overall anatomy of the region. Scale = 100 μm. (**B**) Multichannel in situ hybridization shows Tacr1 transcript (red) is localized to the PBN-SL. Co-staining for the glutamate vesicular transporter Slc17a6 (Vglut2; cyan) and GABA vesicular transporter SLC32A1 (Vgat; green) reveals most Tacr1-positive PBN-SL neurons are glutamatergic and hence excitatory. (**C**) Quantification of the percentage of Tacr1-positive cells co-expressing Tac1 (left plot) and Vglut2 (right plot). (**D**) A cartoon depicting the strategy used to label spinal^Tacr1 presynaptic specializations and PBN-SL^Tacr1 postsynaptic specializations in the same animal. Cre-dependent viral vectors were injected in the lumbar spinal cord (AAV-DIO-SypYFP to label presynaptic termini) and PBN (AAV-PSD95tagRFP to label postsynaptic densities) in Tacr1^Cre mice. (**E**) Example confocal image of a coronal section from the PBN showing a low-magnification view of the organization of Spinal^Tacr1 presynaptic specializations (green) and PBN-SL^Tacr1 postsynaptic specializations (red). Scale = 100 μm.(**F**) Super-resolution imaging (Airyscan) of sections from three different mice showing close apposition of SypYFP and PSD95tagRFP puncta indicative of synaptic connections. Scale = 1 μm. (**G**) Quantification of number of SypYFP puncta with PSD95tagRFP puncta in close apposition (left graph) and vice versa (right graph) demonstrate that the majority of spinal projection neurons target PBN-SL^Tacr1 neurons; n = 7 sections from n = 3 mice. (**H**) A cartoon depicting the viral strategy for anterograde tracing of PBN-SL^Tacr1 neuron projections using injection of AAV-DIO-tdTomato into the

*Figure 2 continued on next page*

*Figure 2 continued*

PBN of Tacr1$^{Cre}$ mice. (I) Confocal image of coronal section showing tdTomato labeling (red) of cell bodies of PBN-SL$^{Tacr1}$ neurons. (J) Dense projections were found in two major brain regions, (left image; boxed regions): the medial thalamus (MTh) and a region encompassing part of the lateral hypothalamus (LH) and the parasubthalamic nucleus (PSTN). Right images show larger magnification views of boxed regions. Scale = 1 mm. The online version of this article includes the following figure supplement(s) for figure 2:

**Figure supplement 1.** Tac1 and Tacr1 are differentially expressed in the PBN.
**Figure supplement 2.** Viral-assisted visualization of PBN-SL$^{Tacr1}$ projection terminals.

the forebrain. Therefore, we anticipated that these neurons may play a specialized role related to pain and next set out to record their responses as well as determine their function.

Tacr1 expression marks a small group of neurons restricted to the PBN-SL. Fiber photometry provides a simple yet sensitive approach for measuring the population responses of genetically defined and anatomically restricted neurons (*Gunaydin et al., 2014*). Therefore, to study how PBN-SL$^{Tacr1}$ neurons are tuned to respond to sensory stimuli, we engineered mice expressing the calcium sensor GCaMP6f (*Chen et al., 2013*) in these neurons and implanted optical fibers to record population-level activity (*Figure 4A,B*). Consistent with our anatomical studies (*Figure 2*), fiber-photometry recordings demonstrated that many types of noxious stimulus strongly activate PBN-SL$^{Tacr1}$ neurons. An example recording (*Figure 4C*) highlights the typical type of long-lasting calcium transient that was elicited by a single pinch of the hindpaw of a lightly anesthetized mouse. Activation of these neurons was observed irrespective of the pinch location (*Figure 4—figure supplement 1A*). Next, we recorded activity in non-anesthetized mice using two types of noxious but non-damaging sustained mechanical stimulation: pinch with blunt forceps (*Figure 4D*) and pinch with the clip assay (*Figure 4E*). Again, PBN-SL$^{Tacr1}$ neurons exhibited robust population calcium responses to this type of stimulation (*Figure 4D,E*). By contrast, punctate mechanical stimuli did not evoke calcium responses in PBN-SL$^{Tacr1}$ neurons (*Figure 4F*). This was surprising since we used stiff von Frey filaments (0.6 g) that reliably cause nocifensive reflexes (*Abdus-Saboor et al., 2019*). Additionally, pinprick also failed to evoke a calcium response (*Figure 4—figure supplement 1B*). Together, these data may indicate that PBN-SL$^{Tacr1}$ neurons preferentially respond to sustained stimuli.

Painful mechanical, thermal, and chemical stimuli are detected by separate nociceptive channels (*Basbaum et al., 2009*; *Gatto et al., 2019*; *Le Pichon and Chesler, 2014*). We next asked whether these pathways converged on PBN-SL$^{Tacr1}$ neurons by recording evoked activity in this population to high temperatures and chemical irritants. In keeping with these neurons playing a general role in pain sensation they exhibited robust activity to hot-plate exposure (*Figure 4G*) and topical application of the pungent component of mustard oil, allyl isothiocyanate (AITC; *Figure 4—figure supplement 2A and B*). Like pinch, both of these are long-lasting painful stimuli that activate distinct sets of peripheral nociceptors (*Basbaum et al., 2009*). However, PBN-SL$^{Tacr1}$ neurons were not activated by short-term noxious heat (Hargreaves test; *Figure 4H*). Interestingly, although AITC is known to cause allodynia and hyperalgesia (*Albin et al., 2008*), this type of skin irritation did not sensitize PBN-SL$^{Tacr1}$ responses to acute touch or heat stimuli (*Figure 4—figure supplement 2C-F*). In combination, the photometry data strongly support PBN-SL$^{Tacr1}$ neurons as playing a major role in an ascending pathway from spinal dorsal horn that is activated by sustained noxious inputs.

We next set out to determine how this circuit influences pain behaviors by generating mice where PBN-SL$^{Tacr1}$ neurons could be chemogenetically activated (*Figure 5A,B*). Although activation of PBN-SL$^{Tacr1}$ neurons did not induce spontaneous directed pain responses, CNO delivery transformed behavior: mice became skittish, avoided handling and repeatedly attempted to escape from the test chambers (possibly a general pain state). This heightened arousal and anxiety were quantified using an open-field arena as increased movement and avoidance of the center (*Figure 5—figure supplement 1A*). Mice now withdrew from the slightest touch with thin von Frey filaments (0.07 g; *Figure 5—figure supplement 1C*), something not seen in saline controls. Moreover, in a sustained pinch assay using a clip applied to the paw, activation of PBN-SL$^{Tacr1}$ neurons increased nocifensive behavior and almost doubled the time a mouse spent shaking the clipped paw (*Figure 5C, D*). Even more strikingly, behavioral changes persisted after the clip was removed (*Figure 5E*). Although saline -treated mice quickly returned to normal resting behaviors, after CNO the same

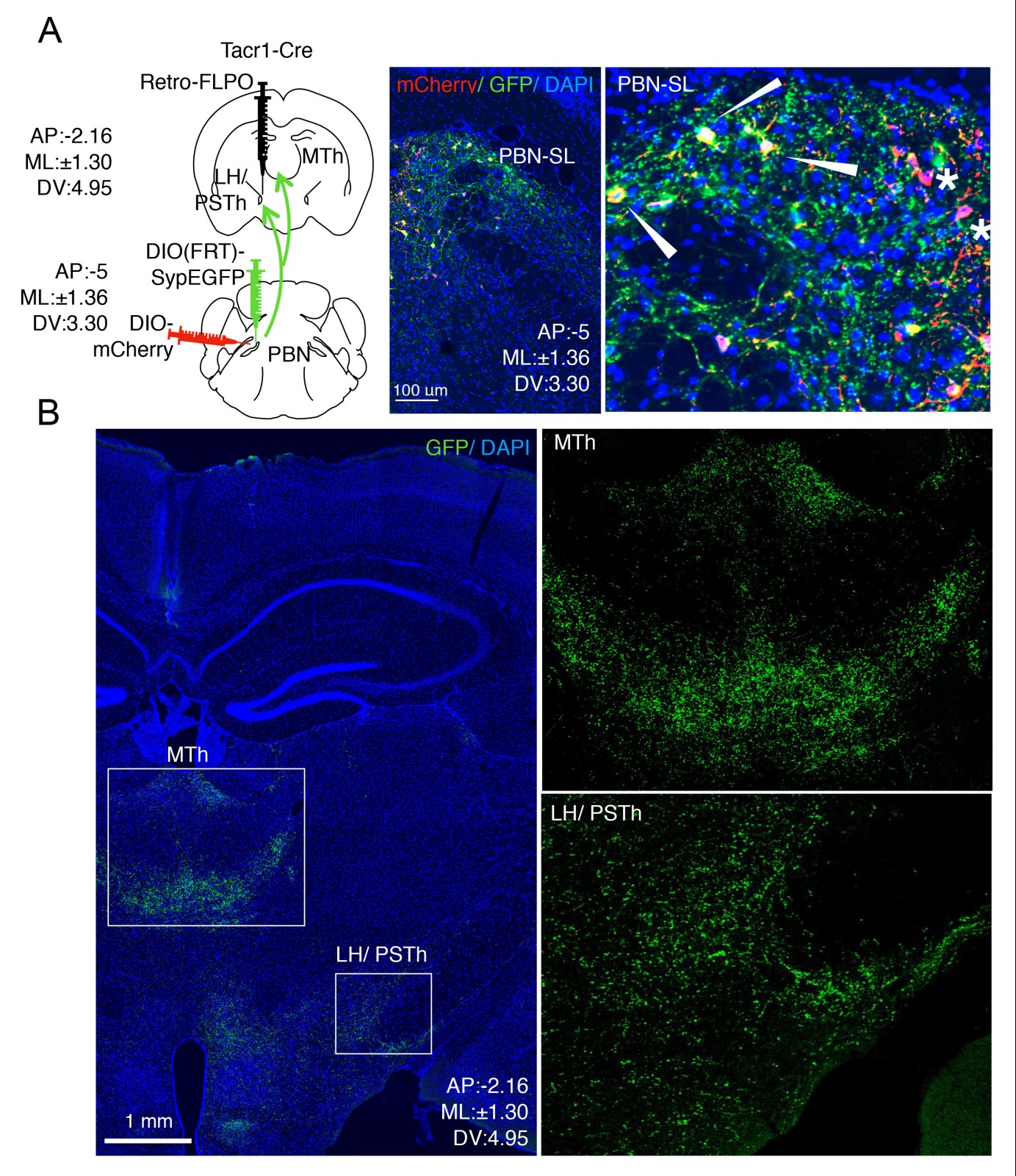

**Figure 3.** PBN-SL[Tacr1] neurons collateralize and project to both MTh and LH/PSTN. (**A**) A complex intersectional genetic strategy was devised to label PBN-SL neurons that project to LH/PSTN and examine if they project to other brain areas (cartoon left side). The terminals of PBN neurons in the LH/ PSTN were transduced with retrograde viral vector encoding Flp recombinase in a Tacr1[cre] mouse. In the same animal, two viral additional vectors were co-injected into the PBN. The first encoded a Flp-dependent presynaptic marker (SypGFP) and the second encoded a Cre-dependent cellular marker

*Figure 3 continued on next page*

*Figure 3 continued*

(mCherry). Under these conditions, the cells bodies and axons of PBN neurons that project to LH/PSTN (and were transduced by the retrograde Flp-virus) are labeled in green and PBN-SL$^{Tacr1}$ neurons are labeled in red. Confocal images of coronal sections of the PBN (right panels) show a high degree of overlap between red (mCherry) and green (SypGFP) confirming the PBN-SL$^{Tacr1}$ neurons project to LH/PSTN (arrowheads). Note that not every cell is co-labelled, as expected when examining the intersecting expression of three different AAVs injected at two distinct sites (stars). Blue = DAPI stain; Scale = 100 μm. (**B**) PBN-SL neurons that project to LH/PSTN have collaterals that make presynaptic terminal specializations in the MTh. Confocal image of a coronal section of a mouse where PBN neurons that project to the LH/PSTN were labeled with SypGFP. As expected, many GFP-positive puncta are seen in the LH/PSTN (boxed region, lower right). Notably, a high density of GFP-positive puncta is also found in the MTh in the same section (boxed region, center). High magnifications of each boxed area are shown on the right side. Blue = DAPI stain; Scale = 1 mm.

animals continued to lick and investigate the hindpaw for many minutes after the pinch was terminated (*Figure 5E*). Equally robust effects of stimulating PBN-SL$^{Tacr1}$ neurons were observed in responses to AITC, a chemical irritant which normally causes licking and attending to the affected area (*Figure 5F*). The time a CNO-treated mouse spent licking the injected site more than doubled. CNO-injected mice also exhibited heightened escape responses when exposed to heat in a hotplate assay (*Figure 5G*). Therefore, this circuit plays a role in controlling the magnitude of behavioral responses to a range of noxious stimuli that activate different peripheral sensory neurons. Interestingly, the latency in a mouse's response to heat, thought to involve simple spinal reflexes, was not influenced indicating that peripheral sensitization is not involved (*Figure 5—figure supplement 1D*). Thus, activation of PBN-SL$^{Tacr1}$ neurons induces a 'hyper-vigilant' state resembling the effects of long-term noxious stimulation (e.g. inflammation) as seen in assays like the formalin test (*Figure 5—figure supplement 1B*). Interestingly, activating the PBN-SL$^{Tacr1}$ neurons using CNO did not enhance responses to formalin, presumably reflecting the potency of this inflammatory stimulus. In a related study, the Sun lab has demonstrated that inhibiting PBN-SL$^{Tacr1}$ neurons reduces behavioral responses to peripheral inflammation (*Deng et al., 2020*).

Recent work from other groups has suggested that inhibiting PBN activity suppresses scratching (*Campos et al., 2018*; *Mu et al., 2017*); however, our data show that stimulating the Tacr1-expressing spinal neurons strongly suppressed itch-related behaviors. Therefore, we next investigated how activation of PBN-SL$^{Tacr1}$ neurons affected behavioral responses to pruritogens. To induce itch, we injected chloroquine, a potent non-histaminergic puritogen that activates a select type of peripheral itch neuron and evokes vigorous scratching (*Han et al., 2013*). Injecting this compound into the nape of the neck of control mice reliably induced localized bouts of scratching starting a few minutes after injection and lasting for about 30 min. Remarkably, chemogenetic stimulation of the small group of PBN-SL$^{Tacr1}$ neurons completely suppressed scratching (*Figure 5—figure supplement 2A and B*), mimicking the effects both of painful stimuli and the fictive pain induced by activating Tacr1-expressing spinal projection neurons. Therefore, the PBN-SL$^{Tacr1}$ neurons serve as a central substrate that is sufficient to suppress itch.

Chemogenetic activation and functional recordings demonstrate that PBN-SL$^{Tacr1}$ neurons respond to painful stimuli and can induce hypervigilance and increased behavioral responses to sustained types of pain. However, it is well known that the central pathways controlling pain sensation and responses involve many regions of the brain (*Bushnell et al., 2013*). Therefore, we next used directed expression of tetanus toxin (TeNT-GFP) (*Han et al., 2015*; *Li et al., 2019*) to silence synaptic output and determine the role of this PBN-SL$^{Tacr1}$ circuit in pain behaviors (*Figure 5I,J*). We again examined responses to different types of sustained stimuli that in control animals induce robust behaviors: the clip assay of extended pinch, and topical application of AITC. Remarkably, in both cases, silencing PBN-SL$^{Tacr1}$ neurons almost completely eliminated shaking and licking normally induced by these stimuli (*Figure 5K–O*). Moreover, at a qualitative level, mice with silenced PBN-SL$^{Tacr1}$ neurons effectively ignored these normally painful stimuli. Thus, this small and localized cluster of PBN-SL$^{Tacr1}$ neurons appears to play a profound role in pain and is necessary for normal behavioral responses.

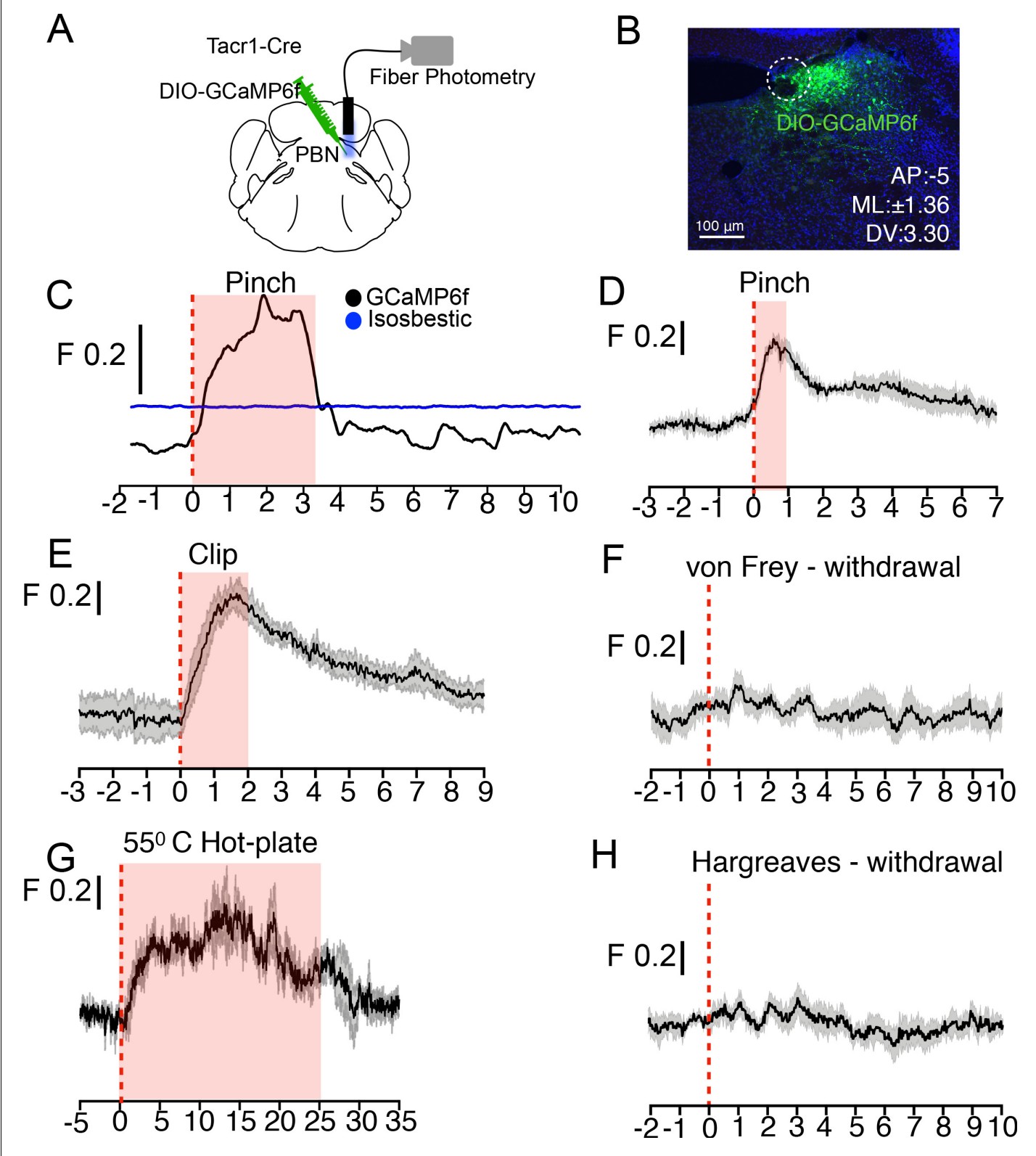

**Figure 4.** PBN-SL[Tacr1] neurons respond to sustained noxious stimulation. (**A**) Cartoon depicting the approach for recording population activity in PBN-SL[Tacr1] neurons: a Cre-dependent viral vector encoding the genetically encoded calcium indicator GCaMP6f was injected into the PBN of Tacr1[Cre] mice and an optical fiber was placed over the injection site. Fiber photometry recording was used to monitor population calcium responses to different somatosensory stimuli. (**B**) After recording, posthoc staining and confocal imaging of PBN sections confirmed GCaMP6f expression (green) and fiber

*Figure 4 continued on next page*

*Figure 4 continued*

placement (dotted line). (**C**) An example photometry trace from a lightly anesthetized mouse showing a robust, time-locked calcium response to tail pinch with blunt forceps (stimulus onset at 0 s; dotted line). Note the response lasts the duration of the stimulation (shaded region) before returning to baseline. As a control for movement artifacts, 405 fluorescence (Blue trace; Isosbestic) was also monitored and showed no changes. (**D–H**) Average population calcium responses in awake mice to sustained (**D, E, and G**) and acute (**F, H**) noxious stimuli. Dark lines are means of responses from multiple animals aligned to the start of stimulation (time 0) with the standard error shown in light grey. X-axis shows time in seconds aligned to the start of stimulation; F indicates ΔF/F scaling (**D**) Pinching (hindpaw) with a blunt forcep (n = 6 mice). (**E**) Clip assay applied to hindpaw (n = 5 mice). (**F**) von Frey stimulation (0.6 g) aligned to paw withdrawal (n = 4 mice) (**G**) 55°C Hot plate test aligned to when mice are placed in a chamber and lasting 25 s (n = 3 mice). (**H**) Radiant heat test (Hargeaves) aligned to paw withdrawal (n = 7 mice). For all traces, time is in seconds, 0 is the start of the trial, change in fluorescence/total fluorescence is shown (black line; **F**).

The online version of this article includes the following figure supplement(s) for figure 4:

**Figure supplement 1.** Population calcium responses in PBN-SL$^{Tacr1}$ neurons to pinch are not somatotopically restricted.

**Figure supplement 2.** The chemical irritant allyl isothiocyanate (AITC) evokes large and sustained increases in calcium signaling in PBN-SL$^{Tacr1}$ neurons.

## Discussion

Together our results identify a circuit involving Tacr1-expressing neurons in the spinal cord and the PBN-SL that controls how mice respond to a wide range of persistently painful stimuli. Interestingly, more than 30 years ago McMahon and Wall described a loop from lamina I dorsal horn to the parabrachial nucleus that influences the activity of lamina I cells (*McMahon and Wall, 1988*). Our results here show that PBN-SL neurons expressing Tacr1 are important for responses to persistent noxious stimuli, exactly as predicted (*Wall et al., 1988*). Importantly, activating the lumbar spinal$^{Tacr1}$ neurons closely mimics persistent pain by inducing pronounced, somatotopically directed behaviors. Very recently and in keeping with our data, other groups have demonstrated that spinal$^{Tacr1}$ neurons are quite diverse (*Sheahan et al., 2020*; *Choi et al., 2020*) and include a population of neurons that selectively innervate the PBN-SL to cause strong nocifensive responses (*Choi et al., 2020*). Interestingly, chemogenetic activation of spinal$^{Tacr1}$ neurons using a different genetic strategy induced itch-related behavior (*Sheahan et al., 2020*). In combination, these data highlight the role of spinal$^{Tacr1}$ neurons in nociception but suggest that the different mouse lines and approaches likely targeted slightly different subsets or numbers of neurons to elicit distinct sensations. Nonetheless our results are entirely concordant with results from the Ginty lab (*Choi et al., 2020*) that implicate projections of spinal$^{Tacr1}$ neurons to a small subnucleus of the PBN as crucial for affective aspects of pain.

We show that a primary target of spinal$^{Tacr1}$ projection neurons are PBN-SL neurons that also express the NKR1-receptor. These PBN-SL$^{Tacr1}$ neurons are both necessary and sufficient to drive a subset of pain-related behaviors. PBN-SL$^{Tacr1}$ neurons are tuned to long-lasting noxious input regardless of location and are selectively important for behaviors that only occur when pain is persistent. When activated, PBN-SL$^{Tacr1}$ neurons dramatically potentiate complex behavioral responses but not simple reflexes. Interestingly, when this group of neurons is stimulated in the absence of noxious peripheral input, mice become 'jumpy' and hypervigilant, now recoiling vigorously from the gentlest of touches that normally would be ignored in the absence of stimulation. For example, response to a first von Frey filament often evoked locomotion and escape from further attempts at stimulation even prior to touch. Comorbidities such as anhedonia and suppression of feeding have been reported to occur during ongoing pain and involve other nuclei in the PBN and their connections with the central amygdala (*Carter et al., 2013*; *Chiang et al., 2020*). Therefore, in the future it will be interesting to determine if PBN-SL$^{Tacr1}$ neurons are also crucial for these types of behavioral changes since PBN-SL$^{Tacr1}$ neurons do not directly target the amygdala but project to distinct regions of the forebrain.

The stimuli that we showed PBN-SL$^{Tacr1}$ neurons respond to are all persistent types of pain. Since both substance P (*Tac1*) and its receptor NKR1 (*Tacr1*) are present both in the spinal projection neurons and in the PBN, it seems likely that neuropeptide signaling may modulate aspects of pain through this pathway. However, given that NKR1-antagonists have limited efficacy in human subjects, we suspect that painful stimulation must normally activate additional pathways that elicit negative valence. Finally, unlike spinal projection neurons, PBN-SL$^{Tacr1}$ neurons do not themselves evoke pain responses or provide positional information but instead appear to drive affective aspects of

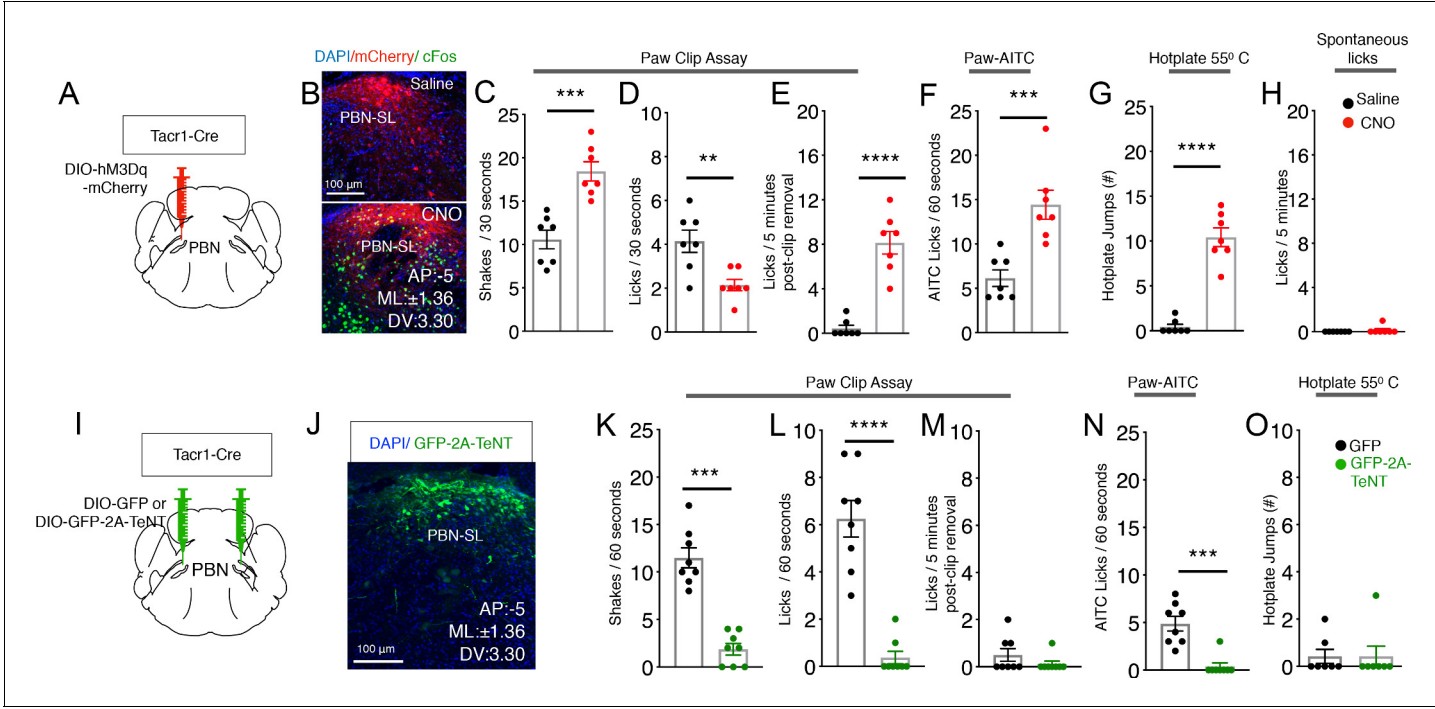

**Figure 5.** PBN-SL[Tacr1] neurons are necessary and sufficient to drive pain-related behaviors. (A–H) Chemogenetic activation of PBN-SL[Tacr1] neurons results in hyperalgesia. (A) A cartoon depicting the strategy for stimulation of PBN-SL[Tacr1] neurons. The PBN of a Tacr1[Cre] mouse was injected with a Cre-dependent viral vector encoding an excitatory DREADD receptor (AAV-DIO-hM3Dq-mCherry) to allow chemogenetic activation. (B) Typical confocal images showing coronal sections of the PBN region from transduced Tacr1[Cre] mice treated with saline (top image) or CNO (bottom image). The expression of mCherry in both animals is restricted to select neurons in the PBN-SL (red). CNO application but not saline injection results in induction of the activity-dependent gene Fos (green) validating the approach. Scale = 100 µm. (C–G) Chemogenetic stimulation of PBN-SL[Tacr1] neurons heightens behavioral responses sustained by noxious mechanical (C–E), chemical (F), and thermal (G) stimulation (n = 7 mice tested with saline, black points and then with CNO, red points on different days). Mice treated with CNO spend significantly more time shaking (C; p=0.0003) and as a result exhibited less licking events (D; p=0.0044) directed to the hindpaw during the Clip assay. Notably, CNO-treated mice continued to lick the hindpaw for several minutes after the clip was removed (E; p=<0.0001). CNO treatment also increased the number of times mice licked their paws after topical AITC treatment (F; p=0.0009) and jumped to escape from a 55°C hotplate (G; p=<0.0001). CNO treatment did not increase result in licking behaviors in the absence of noxious stimulation (H; 0.3370). (I–O) Silencing of PBN-SL[Tacr1] neurons inhibits normal pain responses to sustained noxious stimuli. (I) A cartoon depicting the strategy for silence the output of PBN-SL[Tacr1] neurons. Bilateral injection of a Cre-dependent viral vector encoding Tetanus Toxin Light C GFP fusion (AAV-DIO-GFP-2A-TeNT) in Tacr1[Cre] mice was used to silence PBN-SL[Tacr1] neurons; AAV-DIO-GFP served as a control. (J) Typical confocal image of a coronal section through the PBN region showing the restricted expression of GFP-2A-TeNT in the PBN-SL (green). Scale = 100 µm. (K–O) Silencing of PBN-SL[Tacr1] neuronal output significantly dampened behavioral responses to sustained noxious mechanical (K–M), chemical (N) and thermal (O) stimulation (n = 8 GFP, black points; n = 7 TeNT-GFP, green points). In the Clip (sustained pinch assay, K–M), expression of GFP-2A-TeNT in PBN-SL[Tacr1] neurons decreases the time spent shaking (K; p=0.0003) and the number of licking events (L; p=<0.0001). Similarly, silencing PBN-SL[Tacr1] neurons significantly decreased the number of times mice licked their paws after topical AITC treatment (N; p=0.0001); since control animals did not make escape attempts from a 55°C hot plate (O; p=>0.9999) no effect of neuronal silencing was observed in this assay. Unpaired two-tailed t test; ns > 0.05, ***p≤0.001, ****p≤0.0001.

The online version of this article includes the following figure supplement(s) for figure 5:

**Figure supplement 1.** Activation of PBN-SL[Tacr1] neurons results in a state resembling hyper-vigilance.
**Figure supplement 2.** Activation of PBN-SL[Tacr1] neurons suppresses scratching.

pain. This is particularly apparent in the clip assay where mice with silenced PBN-SL[Tacr1] neurons almost completely ignore this type of sustained, and normally painful, pinch. In the future, determining how other central targets of the spinal[Tacr1] projection neurons control behavior should help reveal if pain responses are simply the sum of distinct parallel circuits or require interactions between distributed regions of the brain.

# Materials and methods

### Key resources table

| Reagent type (species) or resource | Designation | Source or reference | Identifiers | Additional information |
|---|---|---|---|---|
| Genetic reagent (*M. musculus*) | B6;129S-Tac1tm1.1(cre)Hze/J | Jax Mice | Stock number 021877 | Tac1$^{Cre}$ |
| Genetic reagent (*M. musculus*) | Tacr1-T2A-Cre-Neo | Allen Brain Institute | Donated by Dr. Hongkui Zheng | Tacr1$^{Cre}$ |
| Transfected construct (*M. musculus*) | AAV9-CAG-FLEX-tdTomato | UPenn; donated by Allen Institute | Addgene 28306-AAV9 | 200 nl of viral particles (1:1 in saline) at 100 nl/min |
| Transfected construct (*M. musculus*) | AAV9-CAG-FLEX-GFP | UPenn; donated by Allen Institute | Addgene 51502-AAV9 | 200 nl of viral particles (1:1 in saline) at 100 nl/min DRSC. |
| Transfected construct (*M. musculus*) | AAV5-hSyn-FLEX-GCaMP6f | Addgene; donated by Allen Institute | Addgene 100837-AAV5 | 200 nl of viral particles (1:1 in saline) at 100 nl/min |
| Transfected construct (*M. musculus*) | AAV5-hSyn-DIO-mCherry | UNC | Addgene 50459-AAV9 | 200 nl of viral particles (1:1 in saline) at 100 nl/min |
| Transfected construct (*M. musculus*) | AAV5-hSyn-DIO-hM3Dq | UNC; donated by Bryan Roth | Addgene 50474-AAV9 | 200 nl of viral particles (1:1 in saline) at 100 nl/min |
| Transfected construct (*M. musculus*) | AAV8-Flex-hSyn-Synaptophysin-YFP | MGH GDT Core | | 200 nl of viral particles (1:1 in saline) at 100 nl/min |
| Transfected construct (*M. musculus*) | AAV9-Flex-hSyn-PSD95-TagRFP | Chesler Lab, NCCIH | Construct donated by Mark Hoon | 200 nl of viral particles (1:1 in saline) at 100 nl/min |
| Transfected construct (*M. musculus*) | AAVDJ-CMV-eGFP-2A-TeNT | Stanford Viral Core GVVCAAV-71e | GVVCAAV-71e | 200 nl of viral particles (1:1 in saline) at 100 nl/min |
| Transfected construct (*M. musculus*) | AAVRetro-hSyn-FLPo | Levine Lab, NINDS *Sathyamurthy et al., 2020* | | 200 nl of viral particles (1:1 in saline) at 100 nl/min |
| Transfected construct (*M. musculus*) | AAV1-hSyn-FSF-Syp-EGFP | Levine Lab, NINDS *Sathyamurthy et al., 2020* | | 200 nl of viral particles (1:1 in saline) at 100 nl/min |

## Mouse lines

Animal care and experimental procedures were performed in accordance with a protocol approved by the National Institute for Neurological Diseases and Stroke (NINDS) Animal Care and Use Committee. *Tac1$^{Cre}$* mice (Tac1-IRES2-Cre-D or B6;129S-Tac1tm1.1(cre)Hze/J; Stock number 021877) (*Barik et al., 2018*; *Harris et al., 2014*) were purchased from Jackson Laboratories. *Tacr1$^{Cre}$* mice (Tacr1-T2A-Cre-Neo) was kindly donated by Dr Hongkui Zheng, Allen Brain Institute. Genotyping for the mentioned strains was performed according to protocols provided by the Jackson Laboratories.

## Viral vectors and stereotaxic injections

Mice were administered 1 ml saline mixed with 25 mg/kg of ketoprofen 30 min prior to and for 2 days daily post-surgery. Mice were anesthetized with 2% isoflurane/oxygen prior and during the surgery. Craniotomy was performed at the marked point using a hand-held micro-drill (Roboz). A Hamilton syringe (5 or 10 µl) with a glass-pulled needle was used to infuse 200 nl of viral particles (1:1 in saline) at 100 nl/min. The following coordinates were used to introduce the virus: PBN-SL- AP:−5, ML:±1.36; DV:3.30; LH/PSTN- -AP:−1.72, ML:±0.2; DV:3.5. The stereotaxic surgeries to deliver AAVs in the lumbar spinal cord were performed as described before (*Sathyamurthy et al., 2020*). Vectors used and sources: AAV9-CAG-FLEX-tdTomato (UPenn; donated by Allen Institute); AAV9-CAG-FLEX-GFP (UPenn; donated by Allen Institute); AAV5-hSyn-DIO-mCherry (UNC); AAV5-hSyn-DIO-hM3Dq (UNC; donated by Bryan Roth); AAV5-hSyn-FLEX-GCaMP6f (Addgene, donated by Allen Institute); AAV8-Flex-hSyn-Synaptophysin-YFP (MGH GDT Core); AAV9-Flex-hSyn-PSD95-TagRFP (this paper; donated by Mark Hoon); AAVDJ-CMV-eGFP-2A-TeNT (Stanford Viral Core GVVCAAV-71); AAVRetro-hSyn-FLPo (*Sathyamurthy et al., 2020*); AAV1-hSyn-FSF-Syp-EGFP (*Sathyamurthy et al., 2020*). Post hoc histological examination of each injected mouse was used to confirm viral-mediated expression was restricted to target nuclei.

## Behavioral assays

One experimenter carried out all the behavioral assays for the same cohort and was blinded to the treatments (J.T. or M.S.). Experiments throughout all the intraplantar and intraperitoneal administrations were performed by one experimenter (A.B.). All experiments were done in the same room, which was specifically designated for behavior and under red light. Mice were habituated in their home-cages for at least 30 min in the behavior room before experiments. For the Hargreaves test, mice were habituated in transparent plexiglass chambers for 30 min. Clozapine-N-Oxide (1 mg/kg) dissolved in DMSO and diluted in saline was injected i.p. 60–90 min before behavioral experiments or histochemical analysis (*Krashes et al., 2011*). In the assays where spontaneous behaviors were scored, the behaviors were recorded with a digital camera and scored with a stopwatch and a hand-held cell-counter or using the CleverSys software. In the itch assay an individual scratching bout was calculated as a single event. Similarly, for calculating the no. of licks and no. of grooming bouts, each bout was counted as an event. This is because each lick or each scratch by itself is too fast to be reliably scored. Rather, the bout lasting from the start to stop of the behavior was counted as a single event. The shaking behavior did not occur in bouts and thus individual shakes were accounted for during scoring. Hot-plate and Hargreaves tests were scored as described in *Barik et al., 2018*.

The clip assay was performed as described in *Huang et al., 2019*. The experimental mice were extensively handled to be acquainted with the experimenter and before the clip was applied on the hind-paw the mice were physically restrained in hand while the GCaMP signals were being recorded. The mice were restrained on a foam-filter paper bed in a way the hind paw stuck out to enable clip application. While one experimenter restrained the animal, another experimenter applied the clip and took it off. The clip was never kept on the experimental animals for more than 60–70 s. For the AITC test, 10% AITC (Sigma) in saline was injected intradermally in the paw as described in, (*McNamara et al., 2007*). For the formalin test, 2% formalin in saline was injected under the dorsal skin of the right hindpaw with an insulin syringe until a swelling was observed (20ul) (*Barik et al., 2018*). All the experiments were videotaped with an over-head video camera or a panasonic digital camera and scored offline post-hoc using TopScan by CleverSys. The Hargreaves apparatus, and the programmable hotplate were purchased from IITC and used according to the manufacturer's instructions. Von Frey test was done by manually applying the following filaments: 0.008, 0.02, 0.04, 0.07, 0.16, 0.4, 0.6, 1, 1.4 g. Animals received 10 stimulations on the left paw. Inter trial interval was at least 15 s. Open field was from CleverSys. Mice were always habituated in their respective chambers for 30 min prior to experimentation.

## Fiber photometry

A three-channel fiber photometry system from Neurophotometrics was used to collect data. Light from two LEDs (470 nm and 405 nm) were bandpass filtered and passed through a 20 x Nikon Objective focussed on a fiber optic cable coupled to the cannula implanted on mouse PBN-SL. Fluorescence emission was collected through the same patch cord and filtered on a CMOS sensor. Data was acquired through Bonsai. Photometry data was analyzed using a MATLAB code provided by Neurophotometrics. To correct for photobleaching and heat mediated LED decay, the isosbestic signal was fit with a biexponential that was then linearly-scaled to the calcium-dependent fluorescence signal F. The ΔF/F was calculated by dividing the signal by the scaled fit. The start and end of the stimuli (where applicable) were timestamped. Where possible, simultaneous video recording with a Microsoft webcam was performed.

## Immunostaining, multiplex in situ hybridization, and confocal microscopy

Mice were anesthetized with isoflurane and perfused intracardially with PBS and 4% PFA (Electron Microscopy Sciences) consecutively for immunostaining experiments. Fresh brains were harvested for in situ hybridization experiments. Tissue sections were rinsed in PBS and incubated in a blocking buffer (5% goat serum; 0.1% Triton X-100; PBS) for 3 hr at room temperature. Sections were incubated in primary antibodies in the blocking buffer at 4°C overnight. Sections were rinsed 1–2 times with PBS and incubated for 2 hr in Alexa Fluor conjugated goat anti-rabbit/ rat/ chicken secondary antibodies (Thermo Fisher Scientific), washed in PBS, and mounted in ProLong gold mounting media (Thermo Fisher Scientific) onto charged glass slides (Daigger Scientific). Multiplex ISH was done with

a manual RNAscope assay (Advanced Cell Diagnostics). Probes were ordered from the ACD online catalog. Z stack images were collected using a ×20 and ×40 oil objective on a laser scanning confocal system (Olympus Fluoview FV1000) and processed using ImageJ/FIJI software (National Institutes of Health). The AiryScan images of synapses were acquired with 60X oil objective on a Zeiss LSM 880 AiryScan Confocal microscope.

## Statistical analyses

All statistical analyses were performed using GraphPad PRISM seven software. ns >0.05, *p≤0.05, **p≤0.01, ***p≤0.001, ****p≤0.0001.

## Acknowledgements

We are indebted to Nicholas Ryba and Mark Hoon (NIDCR), as well as members of Chesler lab (NCCIH) for invaluable discussions and advice on the manuscript. We are grateful to Hongkui Zheng (Allen Brain Institute) for providing the Tacr1-Cre mice and Mark Hoon for the AAV-DIO-PSD95tagRFP. We thank Dr Carolyn Smith at the NINDS imaging core facility for help with acquiring AiryScan images. This work was supported by the Intramural Research Program of the NIH, National Center for Complementary and Integrative Health (ATC) and National Institute of Neurological Disorders and Stroke (ATC and AL).

## Additional information

### Funding

| Funder | Grant reference number | Author |
|---|---|---|
| National Center for Complementary and Integrative Health | Intramural program | Alexander Chesler |
| National Institute of Neurological Disorders and Stroke | Intramural program | Ariel Levine<br>Alexander Chesler |

The funders had no role in study design, data collection and interpretation, or the decision to submit the work for publication.

### Author contributions

Arnab Barik, Conceptualization, Data curation, Formal analysis, Investigation, Visualization, Methodology, Writing - original draft, Writing - review and editing; Anupama Sathyamurthy, Investigation, Methodology, Writing - review and editing; James Thompson, Mathew Seltzer, Investigation, Methodology; Ariel Levine, Resources, Supervision, Funding acquisition, Writing - review and editing; Alexander Chesler, Conceptualization, Resources, Data curation, Supervision, Funding acquisition, Visualization, Writing - original draft, Project administration, Writing - review and editing

### Author ORCIDs

Arnab Barik  https://orcid.org/0000-0001-6850-0894
Ariel Levine  https://orcid.org/0000-0002-0335-0730
Alexander Chesler  https://orcid.org/0000-0002-3131-0728

### Ethics

Animal experimentation: Animal experimentation: All surgical, experimental and maintenance procedures were carried out in accordance in accordance with a protocol approved by the National Institute for Neurological Diseases and Stroke (NINDS) Animal Care and Use Committee (ASP1365 and ASP1369).

### Decision letter and Author response

Decision letter https://doi.org/10.7554/eLife.61135.sa1
Author response https://doi.org/10.7554/eLife.61135.sa2

## Additional files

### Supplementary files

- Source data 1. Source data for all figures.

- Transparent reporting form

### Data availability

All data generated or analysed during this study are included in the manuscript and supporting files. Source data have been uploaded.

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
