## [Decision Letter]

**Acceptance summary:**

The authors of this paper show that chemogenetic activation of spinal cord neurons that expression the Tacr1 receptor, which responds to substance P, evokes many nocifensive behaviors that correlate with the experience of pain. The effects are exerted via connections with the parabrachial nucleus of the brainstem, which in turns connects with brain circuits that underlie emotional components of the pain experience.

**Decision letter after peer review:**

Thank you for submitting your article "A spinoparabrachial circuit defined by Tacr1 expression drives pain" for consideration by *eLife*. Your article has been reviewed by three peer reviewers, one of whom is a member of our Board of Reviewing Editors, and the evaluation has been overseen by Richard Aldrich as the Senior Editor. The following individual involved in review of your submission has agreed to reveal their identity: Richard D Palmiter (Reviewer #3).

The reviewers have discussed the reviews with one another and the Reviewing Editor has drafted this decision to help you prepare a revised submission.

Summary:

This is a concise paper describing a projection from spinal cord to Tacr1-expressing neurons in the PBN. The authors conclude that activation of this spinal cord > PBN Tacr1 circuit "does not trigger pain responses but instead serves to dramatically heighten nocifensive behaviors and suppress itch" and that mice with "silenced Tacr1 neurons ignore long-lasting noxious stimuli”. These are interesting and important findings that complement a recent paper (Deng et al., 2020). The conclusions reached by the authors are generally substantiated, but considerable revision is required, but some additional experiments are essential. These experiments will not only strengthen the authors' conclusions, but may provide some insight as to underlying mechanism as to the very interesting finding relating pain to itch inhibition.

Essential revisions:

Major concerns several requiring additional experimental information:

1) This paper appears to have been written for a different journal. As the authors have more space in *eLife* they should move some of the more interesting supplementary figures to main figures. They should also expand on the Discussion and take advantage of the Deng/Sun paper to compare results of the two studies to create an integrated picture.

2) The Fos induction after activating Tacr1 neurons in spinal cord is not impressive and does not appear to be colocalized with red cells expressing the DREADD (Figure 1—figure supplement 4C). Quantification Fos overlap with Tacr1 and Fos in non-Tacr1 cells is needed. Likewise, quantification of overlap of Tacr1 and Tac1 in lamina 1 of spinal cord is needed.

3) A poke (von Frey) and a pinch (squeeze) are different, so the conclusion that the PBN Tacr1 neurons only respond to sustained stimuli needs better validation. A revision should definitely include a different painful stimulus; foot shocks that may be easier to precisely time. Alternatively a dry ice application test could be used.

4) The authors say that "PBN-Tacr1 neurons induces a "hyper-vigilant" state resembling the effects of long-term noxious stimulation (e.g. inflammation)." Which raises the question of how the mice respond to an inflammatory stimulus (e.g. carrageenan or formalin)?

5) Most importantly, what percentage of spinal Tacr1-expressing neurons are projection neurons? The authors gave the impression that Tacr1-expressing cells in spinal cord are all projection neurons, but as Todd has reported some are undoubtedly interneurons. This should be quantified.

6) The question of the nature of the NK1R neurons is particularly relevant given the recent contribution of the NK1R neurons from the Sarah Ross lab, which has a Biorxiv paper indicating NK1R(Tacr1)-expressing spinal interneurons are involved in itch (https://www.biorxiv.org/content/10.1101/2020.07.14.199471v1.full). Of course, this paper indicated that activating Tacr1-neurons suppresses itch. This should be discussed.

7) Here also is where an additional experiment could not only focus on the difference in the results from the Ross lab, but also provide some insight into mechanism:

8) The inhibition of nape of the neck chloroquine-induced scratching is perhaps the most remarkable and exciting finding. Does the mechanism involve inhibition at the level of the spinal cord (via an ascending-descending loop), or is it an effect that occurs supraspinally? If the latter, then one would predict that nape of the neck chloroquine-evoked cervical cord Fos expression would persist after lumbar cord DREADD stimulation. This is an easy experiment, somewhat complicated perhaps by the fact that the loss of scratching would inevitably reduce Fos in the cervical cord. But any Fos attributed to the chloroquine activation of pruritoceptors should either persist, if the inhibition occurs supraspinally, or disappear, if the inhibition occurs at the level of the cord. This is an experiment that should absolutely be included in a revision.

9) Lastly, it would great if the authors could provide videos of activating Tacr1 neurons showing pain like behaviors.

---

## [Author Response]

Essential revisions:Major concerns several requiring additional experimental information:1) This paper appears to have been written for a different journal. As the authors have more space in eLife they should move some of the more interesting supplementary figures to main figures. They should also expand on the Discussion and take advantage of the Deng/Sun paper to compare results of the two studies to create an integrated picture.

We have rewritten the paper and included multiple citations and discussion of all requested literature.

2) The Fos induction after activating Tacr1 neurons in spinal cord is not impressive and does not appear to be colocalized with red cells expressing the DREADD (Figure 1—figure supplement 4C). Quantification Fos overlap with Tacr1 and Fos in non-Tacr1 cells is needed. Likewise, quantification of overlap of Tacr1 and Tac1 in lamina 1 of spinal cord is needed.

New Figure 1—figure supplement 1 now more explicitly shows the correspondence; we have also added quantification as requested.

3) A poke (von Frey) and a pinch (squeeze) are different, so the conclusion that the PBN Tacr1 neurons only respond to sustained stimuli needs better validation. A revision should definitely include a different painful stimulus; foot shocks that may be easier to precisely time. Alternatively a dry ice application test could be used.

We added data for additional stimuli including pinprick to which mice showed no Ca-response in these neurons (Figure 4—figure supplement 1B); similarly dry ice did not evoke Ca-transients.

4) The authors say that "PBN-Tacr1 neurons induces a "hyper-vigilant" state resembling the effects of long-term noxious stimulation (e.g. inflammation)." Which raises the question of how the mice respond to an inflammatory stimulus (e.g. carrageenan or formalin)?

We added the formalin test (Figure 5—figure supplement 1) and observed no enhanced effect. I.e. these are not additive responses likely because the activation of Tacr1-neurons already reaches a ceiling. We also reference the Sun paper where they used inhibition of the neurons and reduced behavioral responses to inflammation.

5) Most importantly, what percentage of spinal Tacr1-expressing neurons are projection neurons? The authors gave the impression that Tacr1-expressing cells in spinal cord are all projection neurons, but as Todd has reported some are undoubtedly interneurons. This should be quantified.We used standard methods in the field (Gramdi et al., 2009, see also the recent Ross paper on these neurons) to quantify the presumptive projection neurons based on soma size (see Figure 1—figure supplement 2).6) The question of the nature of the NK1R neurons is particularly relevant given the recent contribution of the NK1R neurons from the Sarah Ross lab, which has a Biorxiv paper indicating NK1R(Tacr1)-expressing spinal interneurons are involved in itch (https://www.biorxiv.org/content/10.1101/2020.07.14.199471v1.full). Of course, this paper indicated that activating Tacr1-neurons suppresses itch. This should be discussed.

We added substantial discussion of this study along with recent work from the Ginty lab that is also related to our findings. Together these papers help clarify the importance of these projection neurons and their heterogeneity (in part addressing point 7)

7) Here also is where an additional experiment could not only focus on the difference in the results from the Ross lab, but also provide some insight into mechanism:

See points 6 and 7

8) The inhibition of nape of the neck chloroquine-induced scratching is perhaps the most remarkable and exciting finding. Does the mechanism involve inhibition at the level of the spinal cord (via an ascending-descending loop), or is it an effect that occurs supraspinally? If the latter, then one would predict that nape of the neck chloroquine-evoked cervical cord Fos expression would persist after lumbar cord DREADD stimulation. This is an easy experiment, somewhat complicated perhaps by the fact that the loss of scratching would inevitably reduce Fos in the cervical cord. But any Fos attributed to the chloroquine activation of pruritoceptors should either persist, if the inhibition occurs supraspinally, or disappear, if the inhibition occurs at the level of the cord. This is an experiment that should absolutely be included in a revision.

New Figure 1—figure supplement 3 directly addresses this point and shows that there is robust itch-evoked cervical Fos induction even when lumbar Tacr1-neurons are activated. Thus the inhibition of scratch is super-spinal again speaking to mechanism (point 7).

9) Lastly, it would great if the authors could provide videos of activating Tacr1 neurons showing pain like behaviors.

Now included.